



# Has the 2022 Hunga eruption impacted the noctilucent cloud season in 2023/24 and 2024?

Sandra Wallis[1], Matthew DeLand[2], and Christian von Savigny[1]

[1]Institute of Physics, University of Greifswald, Felix-Hausdorff-Str. 6, 17489 Greifswald, Germany
[2]Science Systems and Applications Inc., Lanham, Maryland, USA

**Correspondence:** Sandra Wallis (sandra.wallis@uni-greifswald.de)

**Abstract.**

The 2022 Hunga Tonga - Hunga Ha'apai eruption emitted approximately 150 Tg $H_2O$ into the middle atmosphere which is still detectable two years after the event. Microwave Limb Sounder (MLS) observations show that the Hunga $H_2O$ reached the upper polar mesosphere in the Southern Hemisphere in the beginning of 2024, increasing the $H_2O$ mixing ratio in January by about 1 ppmv between 70°S - 80°S up to an altitude of 83 km. No clear signal was detected for the noctilucent cloud occurrence frequency inferred from Ozone Mapping and Profiling Suite - Limb Profiler (OMPS-LP) measurements. It cannot, however, be ruled out that a slight increase from mid-January to February is potentially caused by the additional water vapour from the Hunga event. Several months later, the water vapour anomaly reached the polar summer mesopause region in the NH during the 2024 NLC season. However, a subsequent anomalous warming during the second half of the season might have hindered the ice particle formation, leading to a decrease in occurrence frequency of the mesospheric clouds compared to previous years. To summarise, the volcanic water vapour seems to need two years to reach the summer polar mesopause region. This resembles the Krakatau case that is argued to have caused the first sightings of noctilucent clouds two years after its eruption in 1883.

## 1 Introduction

Back in August 1883, the Krakatau volcano was torn apart by a phreatomagmatic eruption with a volcanic explosivity index (VEI) of 6 (Oemaiya and Santoso, 2019) that released a potentially massive amount of $H_2O$ into the middle atmosphere. Estimates of the injected $H_2O$ mass, however, remain uncertain. A model study by Joshi and Jones (2009) assumed an injected mass of 500 Tg $H_2O$, whereas Thomas et al. (1989) assumed a mass between 100 to 200 Tg $H_2O$. The ash plume was clearly visible to nearby eyewitnesses who reported column heights up to 40 km (Self, 1992). These reports make it plausible that the emitted and entrained $H_2O$ from the Krakatau eruption could have reached the middle to upper stratosphere as well. This event caused optical phenomena, e.g. green twilight skies (von Savigny et al., 2024), blue suns/moons (Wullenweber et al., 2021), Bishop's rings (Kiessling, 1885) and afterglows (Symons, 1888) that were seen and reported in Europe and North America over the following weeks and months.

Two years later, in June 1885, first sightings of noctilucent clouds were reported (Backhouse, 1885; Leslie, 1885; Schröder, 1999). These are mesospheric clouds made of ice particles with radii generally smaller than about 100 nm that are located




in the mesopause summer region at approximately 83 km altitude. They are visible as silvery translucent clouds against the dark twilight sky and best seen from the ground at mid- to high latitudes when the sun is 5° - 15° below the horizon. These clouds form under specific conditions, i.e. mesospheric temperatures lower than about 150 K as well as the presence of $H_2O$ and nucleation nuclei (Rapp and Thomas, 2006; von Savigny et al., 2020).

Some studies suggest that the additional water vapour and possibly nucleation nuclei from the Krakatau eruption resulted in a boost of cloud brightness making them finally visible to the bare eyes two years after the eruption (Thomas et al., 1989). They argue that no convincing reports of twilight phenomena resembling noctilucent clouds exist before 1885 and that skilled observers were present (Schröder, 1999). A causality between the Krakatau eruption and the first appearance of noctilucent clouds two years later is, however, still under debate in the community.

Recently, a phreatomagmatic eruption of the Hunga volcano (VEI of approximatly 6 (Poli and Shapiro, 2022)) on January 15, 2022 opened up this discussion again. Approximately 150 Tg $H_2O$ were emitted into the middle atmosphere (Millan et al., 2022), reaching altitudes up to 55 or even 57 km (Carr et al., 2022). After an initial phase of subsidence in the first two weeks, the tropical Hunga $H_2O$ plume remained at pressure levels between 20 - 40 hPa until it started to rise in October 2022 and reached the tropical stratopause in March 2023 (Niemeier et al., 2023). As the additional water vapour entered the mesosphere it followed the large scale vertical and meridional transport in the middle atmosphere. This study investigates whether the additional water vapour from the Hunga eruptions affected the mesospheric noctilucent clouds (NLCs) during the Southern Hemispheric (SH) NLC season 2023/2024 and during the Northern Hemispheric (NH) 2024 season. It therefore tries to answer the question: could the Hunga eruption trigger a similar impact on the noctilucent clouds two years after the eruption occurred, analogous to the observations after the Krakatau event?

We use Microwave Limb Sounder (MLS) $H_2O$ mixing ratio observations as well as data from the Ozone Mapping Profiling Suite - Limb Profiler (OMPS-LP) to investigate the SH NLC season in 2023/24 and the NH 2024 season. This paper is structured as follows. Section 2 introduces the MLS and OMPS-LP data sets and describes their analysis. Section 3 presents the results of this study that are discussed in Section 4. This paper concludes with Section 5.

## 2  Data analysis

### 2.1  Microwave Limb Sounder (MLS)

We use level 2 version 5 $H_2O$ mixing ratios, temperatures, ice water content and geopotential height from the NASA Microwave Limb Sounder (MLS) to observe the transport of water vapour through the middle atmosphere and the ambient conditions in the NLC region (Waters et al., 2006). MLS is a limb sounding instrument that measures in the microwave spectral region. It was launched onboard the Aura satellite that is placed in a near-polar, sun-synchronous orbit with an inclination of 98°. This allows measurements from 82°N to 82°S with a vertical resolution of 1.3 - 3.6 km between 316 and 0.22 hPa and 6 - 11 km above 0.22 hPa for the $H_2O$ product. The accuracy is 5 - 35% and the precision is 5 - 16% between 316 and 1 hPa (Livesey et al., 2022). Only mixing ratios up to 0.001 hPa are recommended for scientific use.





The data was filtered according to the data documentation (Livesey et al., 2022). Only $H_2O$ profiles that are associated with a positive precision, a status field with an even number, a quality flag greater than 0.7 and a convergence less than 2.0 are used. Profiles with mixing ratios less than 0.101 ppmv at altitudes $\leq$ 1 hPa were dismissed. The temperature data was analyzed

similarly, but only profiles with a quality flag greater than 0.2 at pressures $\leq$ 83 hPa and with a quality flag greater than 0.9 at pressures $\geq$ 100 hPa were used. The convergence needed to be less than 1.03 whereas profiles between 261 - 100 hPa with a 215 hPa ice water content greater than 0.005 $g/m^3$ were omitted. Several profiles between 261 - 178 hPa were rejected due to criteria described in Livesey et al. (2022).

Geopotential height, also provided by MLS, is utilized to interpolate the $H_2O$ mixing ratios from a pressure grid to geometric

altitudes (z) using the following equation (Guinn and Mosher, 2015):

$$z = \frac{r \cdot H}{r - H} \tag{1}$$

Here, H is the MLS geopotential height and r the mean Earth radius (6371000 m). Afterwards, daily zonal mean profiles for $10°$ and $5°$ latitude bins are determined, respectively. Water vapour anomalies are calculated by subtracting the multi-annual mean for each day determined from the reference period from 2017 - 2021. Due to the strong volcanic signal, only $H_2O$ mixing

ratio anomalies larger than 3 times the standard deviation of the reference period are considered significant.

## 2.2    Ozone Mapping and Profiling Suite - Limb Profiler (OMPS-LP)

The Ozone Mapping and Profiling Suite - Limb Profiler (OMPS-LP) instrument measures limb-scattered sunlight and was launched onboard of the sun-synchronous Suomi National Polar-orbiting Partnership (SNPP) satellite. Its hyperspectral measurements between 290 - 1000 nm cover an altitude range from the surface to 80 - 100 km with a vertical resolution of 1.6

km (Jaross et al., 2014). NLCs can be detected in the OMPS-LP limb radiance profile, because they lead to a radiance enhancement in the upper mesosphere. Individual radiance profiles at 353 nm are compared to a zonal mean radiance profile from mid-latitudes (where no NLCs are expected), and the maximum value of this radiance ratio is identified. The detection method is described in DeLand and Gorkavyi (2021).

All profiles were zonally and daily binned with a latitude bin size of $5°$. Only altitudes between 80 and less than 90 km

are considered and a threshold of 2.0 is applied to the maximum radiance ratio. The noctilucent cloud occurrence frequency is calculated by determining the ratio of cloud events after the filtering process and the number of all measurements times 100%. Finally, the occurrence frequency is smoothed by a 5 day running mean.

## 3    Results

### 3.1    The SH 2023/24 NLC season

The additional water vapour emitted from the Hunga eruption in January 2022 can be tracked as an anomaly using the MLS instrument (e.g. Niemeier et al. (2023)). Over the period of two years it spread zonally, vertically and meridionally (Nedoluha



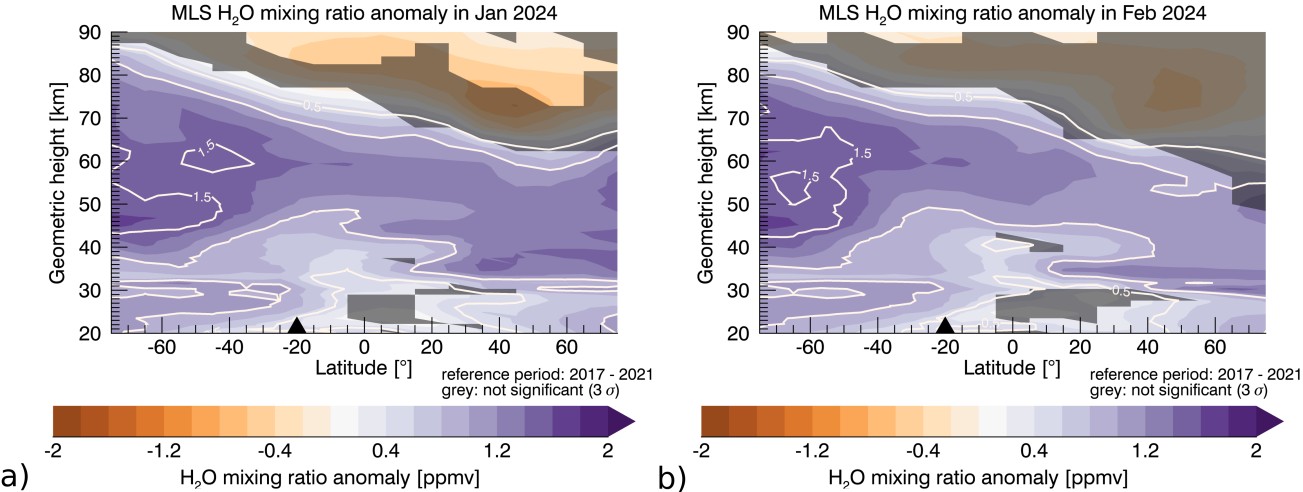

**Figure 1.** Monthly mean water vapour mixing ratio anomaly from MLS data for a) January and b) February 2024. The reference period is 2017 - 2021 and anomalies that are less than 3 standard deviations of the reference period are considered not significant and marked with a dark grey colour. The triangle denotes the latitude position of the Hunga volcano.

et al., 2024). Figure 1a shows the latitudinal and vertical distribution of the zonally averaged and monthly mean water vapour anomaly in January 2024 corresponding to a reference period of 2017 - 2021, i.e. five years before the Hunga eruption. Areas are shaded in grey where the $H_2O$ mixing ratio anomalies do not exceed 3 times the standard deviation within the reference period and are thus considered insignificant. The water vapour anomaly in the mid-latitude NH is located up to 60 km in January. In the tropics, regions up to 80 km are significant whereas in the summer polar region (specifically between 70°S - 80°S), the water vapour anomaly is significant from the lower stratosphere up to the upper mesosphere, reaching 88 km. The water vapour anomaly often exceeds about 1.5 ppmv in the SH between 80°S - 30°S and 40 - 65 km. Anomalies up to 1 ppmv reach 83 km between 70°S - 80°S. The water vapour anomaly mixing ratios increase in February for the SH (Figure 1b), where mixing ratio anomalies of 1.5 ppmv are found up to 75 km altitude for the highest SH latitudes. Similar plots for all months of 2022 to 2024 are shown in Figure S1 to Figure S3 in the supplementary information.

Noctilucent clouds can be described using different variables derived from observations, one of which is the occurrence frequency. This measure provides the percentage of cloud detection compared to the number of total observations. We use the maximum radiance ratio between the measured and background signal of each OMPS-LP profile to calculate the NLC occurrence frequency for each SH season since the SNPP satellite launch. Figure 2a compares the occurrence frequency for 2023/24 (red line) to the previous seasons as well as the average of the seasons 2016/17 to 2020/21 ± one and three standard deviation intervals, respectively. The start of the NLC season varies between different years and results in a large variability. This is due to a variable date for the change from winter to summer circulation in the stratosphere (e.g. due to a late breakdown of the polar winter vortex) that impacts the gravity wave filtering in the summer hemisphere, the meridional circulation, the





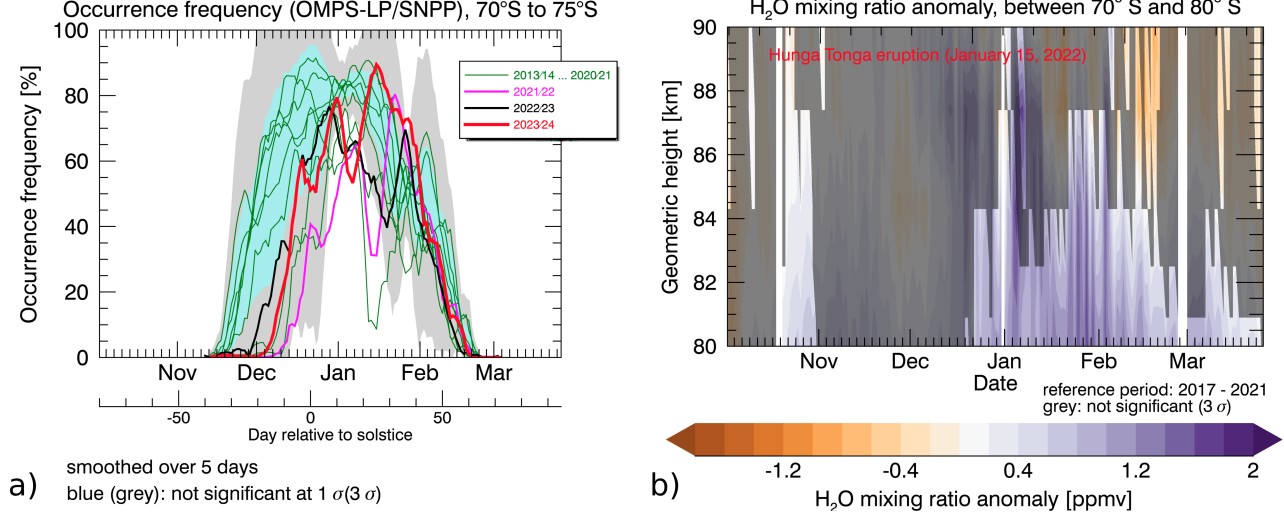

**Figure 2.** a) NLC occurrence frequency from OMPS-LP between 70°S - 75°S. The thick lines indicates the occurrence frequency for the SH 2021/22 (magenta), 2022/23 (black) and 2023/24 (red) seasons, respectively. All seasons before are displayed as thin green lines. The blue (grey) shaded region contains the averaged occurrence frequency for the years 2016/17 to 2020/21 ± one (three) standard deviations from this period. The day of solstice is December 21. b) MLS water vapour anomaly between 70°S - 80°S for altitudes between 80 to 90 km during the SH NLC season. Grey areas are considered not significant (less than 3 $\sigma$).

temperature in the upper mesosphere and hence the conditions for noctilucent clouds (Karlsson et al., 2011). The occurrence frequency from mid-January to February 2024 is higher than one standard deviation of the average previous seasons (compare with the pale blue shading) and even shortly exceeds the 3 $\sigma$ interval at the end of February. Occurrence frequencies for other latitudes (85°S - 65°S) are depicted in the supporting information (Figure S4). We hypothesize that the additional water vapour from Hunga might have had an impact on the NLC properties. Therefore, Figure 2b shows the $H_2O$ mixing ratio anomaly

between 80 - 90 km at 70°S - 80°S. Significant anomalies appear from mid-October to November (before the NLC season started) and from mid-December to the end of March, where they shortly reach altitudes up to 90 km at the beginning of January and 87 km at its end.

    The properties of noctilucent clouds depend on both the mesospheric temperature and the amount of water vapour. Figure 3a compares the temperature at NLC altitude for the 2023/24 season with previous seasons as well as the intervals of the averaged

2016/17 to 2020/21 seasons ± one and three standard deviations, respectively. The NLC altitude was determined from OMPS-LP measurements. It is clear that the temperature in 2023/24 is similar to the previous temperature time series. Figure 3b shows a similar comparison for the $H_2O$ mixing ratios at NLC altitude. The $H_2O$ mixing ratio after January 2024 is larger by up to 1 ppmv compared to the previous years. Comparisons for more latitude bins between 80° - 65° are shown in the supporting information (Figure S5).





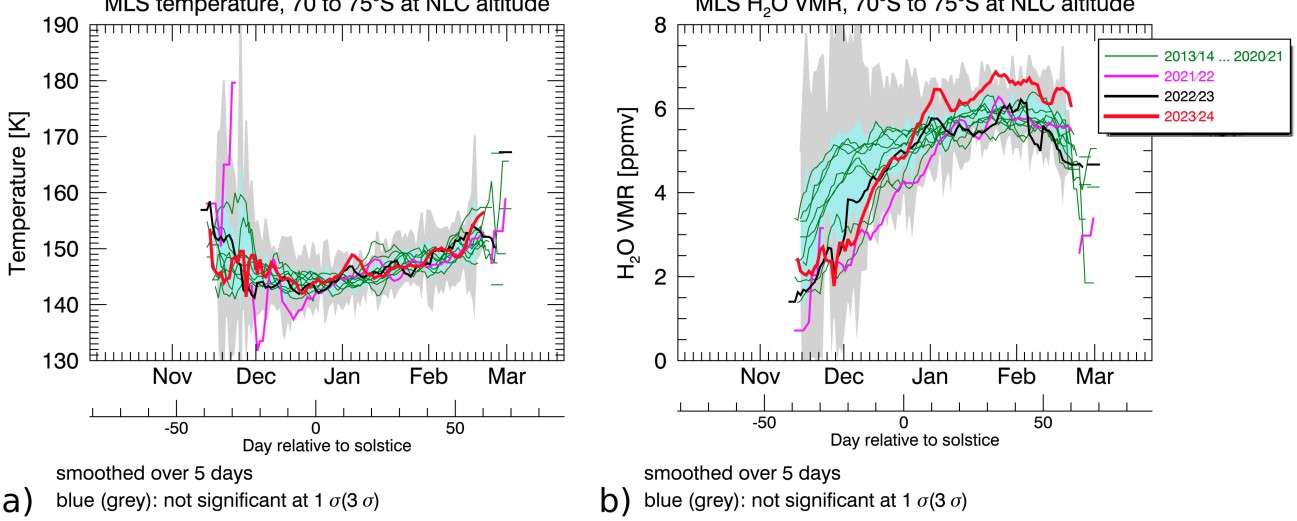

**Figure 3.** a) MLS temperature between 70°S - 75°S at NLC altitude determined from OMPS-LP measurements. The thick lines show the SH 2021/22 (magenta), 2022/23 (black) and 2023/24 (red) seasons, respectively. All seasons before are displayed as thin green lines. Blue (grey) areas indicate the averaged temperature from 2016/17 to 2020/21 ± one (three) standard deviations. b) The same for MLS water vapour mixing ratios.

Figure 4 shows the maximum radiance ratio between the measured and background limb radiance from OMPS-LP measurements between 70°S to 75°S using a similar color scheme as in Figure 2a to indicate the NLC seasons and the average intervals. The maximum radiance ratio for 2023/24 is greater from mid- to end of January than in all years prior, exceeding the $3\,\sigma$ interval of the reference period for a short time. Comparisons for other latitude bins are presented in Figure S8.

The altitudes of the NLCs were determined using OMPS-LP measurements and are shown in Figure 5 for the latitude band between 70°S - 75°S. In general, the NLCs are detected at the highest altitudes at the beginning of the season before they tend to descend, reaching the minimum altitude in the beginning of February (Bailey et al., 2005). Afterwards, there is a strong variability in the NLC altitude until the end of the season. Comparisons of the different season for latitudes between 85°S to 65°S are shown in the supporting information (Figure S6).

We also created scatter plots between MLS $H_2O$ and MLS temperature at NLC altitudes for 65°S - 70°S, 70°S - 75°S and 75°S - 80°S, respectively (Figure 6a-b). Data points from the seasons 2013/14 to 2022/23 are depicted as empty circles whereas the measurements for 2023/24 are highlighted with filled circles, using a red colour for measurements before January 15 and a green colour afterwards. All three chosen latitude bands show a clustering of filled circles for high $H_2O$ with the highest mixing ratios reached after January 15.

We further hypothesize that the additional water vapour could increase the NLC occurrence frequency at a specific mesospheric temperature compared to an unperturbed case. Figure 6d-f shows the scatter plots for OMPS-LP occurrence frequency



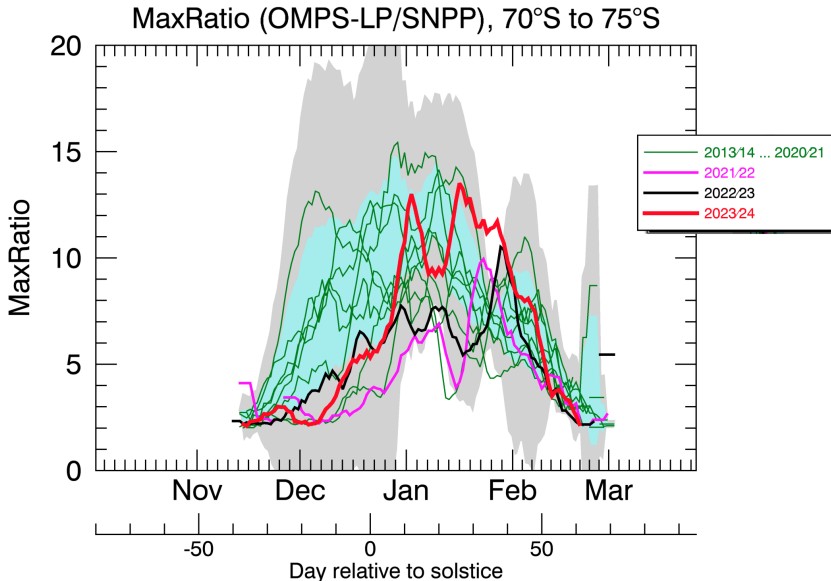

**Figure 4.** The maximum radiance ratio between the measured and background limb radiance from OMPS-LP for each of the SH NLC seasons. The thick lines indicate zonally and daily mean maximum radiance ratios between 70 °S to 75 °S for 2021/22 (magenta), 2022/2023 (black) and 2023/24 (red). All seasons before are displayed as thin green lines. The blue (grey) shaded region contains the averaged maximum radiance ratio for the years 2016/17 to 2020/21 $\pm$ one (three) standard deviations from this period. The day of solstice is December 21.

and MLS temperature at NLC altitude similar to Figure 6a-c. The scatter plots at the three highest SH latitudes indicate that the occurrence frequency is slightly higher in 2023/24 for a specific temperature than in previous years especially after January 15. The correlation between the temperature at NLC altitude and the NLC occurrence frequency is, however, weak. Similar scatter plots for latitudes 60°S - 65°S and 65°S - 70°S are shown in Figure S7 where the occurrence frequency during the core
season is not as high, making a proposed effect of the increased water vapour potentially clearer. No distinct impact of the water vapour is, however, seen at these latitudes.

## 3.2 The NH 2024 NLC season

Similar to the SH 2023/24 season, the $H_2O$ anomaly from the eruption is also detected up to the NLC altitude in mid-2024 (see Figure S3) and could therefore potentially affect the mesospheric clouds, making the NH 2024 NLC season relevant to
the Hunga discussion. Monitoring the $H_2O$ distribution during that season is, however, much more difficult, because the MLS $H_2O$ channel is only partially active since April 29 in order to extend the instrument's lifetime. Figure 7a shows the occurrence frequencies between 60 - 65°N for each year, highlighting again 2024 with a red line. The 2024 season exhibits a peak in mid-



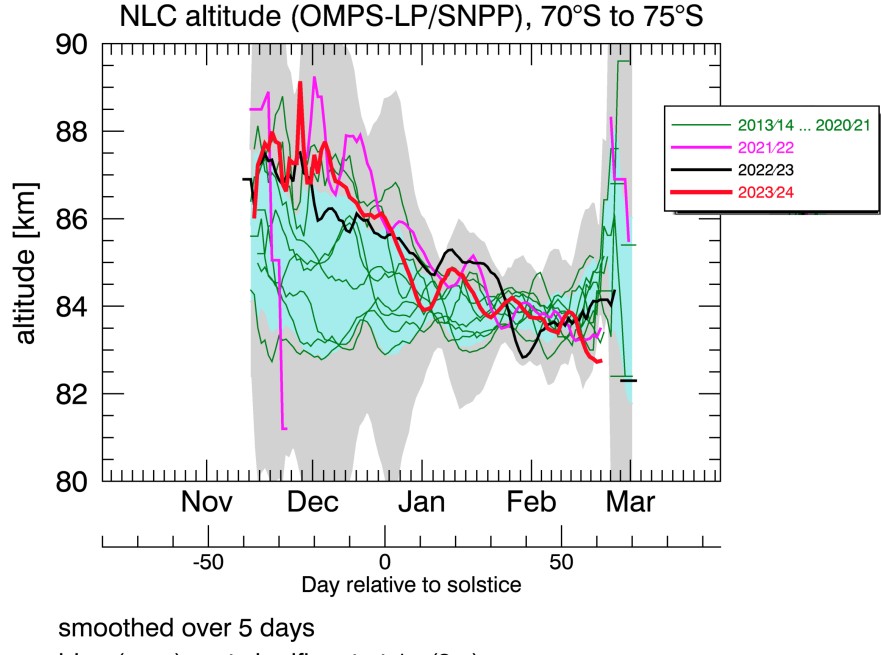

**Figure 5.** Altitude of the NLCs determined from OMPS-LP measurements. The thick lines indicate zonally and daily mean maximum radiance ratios between 70 °S to 75°S for 2021/22 (magenta), 2022/23 (black) and 2023/24 (red). All seasons before are displayed as thin green lines. Blue (grey) areas indicate the mean NLC altitude from 2016/17 to 2020/21 ± one (three) standard deviation.

July (60 %) that exceeds 1 standard deviation ($\sigma$) of the averaged reference period. Comparing this to the sparse data from the $H_2O$ channel, there is a positive $H_2O$ anomaly of nearly 1 ppmv in mid-July. The occurrence frequency rapidly decreases after
the July peak and is lower than the 1 $\sigma$ interval for most of the remaining 2024 season. During this second half of the season, MLS detected a positive temperature anomaly of up to about 10 K compared to the mean average. Unfortunately, MLS has a limited vertical resolution at these altitudes. For a better context, Figure 7d-e shows MLS $H_2O$ and temperature anomalies together with a significance estimate as well as the NLC altitude from OMPS-LP measurements.

Further at higher latitudes, occurrence frequencies of the 2024 season between 65 - 80°N are highest at the end of June
and the beginning of July where they partially exceed the 1 $\sigma$ interval (Figure 8a,d,g). Again, they decrease during the second half of the season and are lower than the 1 $\sigma$ interval - even shortly declining below the 3 $\sigma$ area (mid-August in Figure 8g). MLS $H_2O$ mixing ratios exceed the 1 $\sigma$ interval and are partially higher than 3 $\sigma$ during the mid-July measurement window (Figure 8c,f,i). An increase of MLS temperature during the second half of the season is also observed at these latitudes.





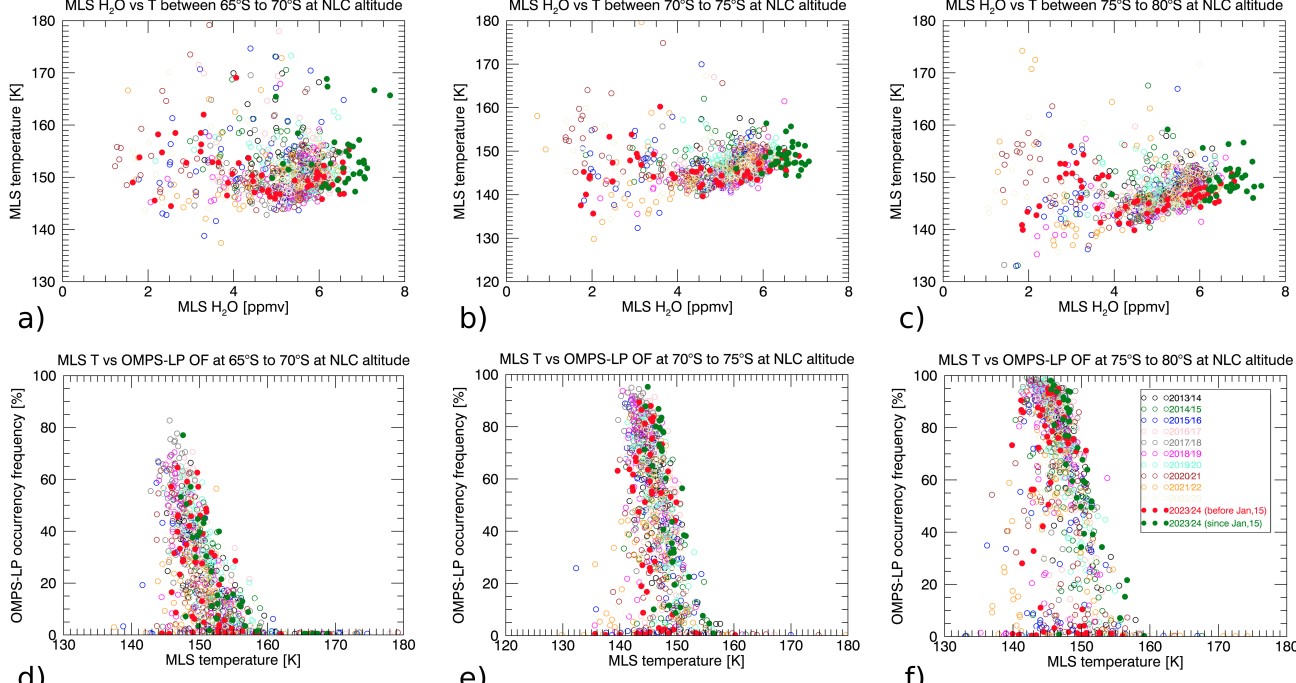

**Figure 6.** Scatterplot for MLS temperature versus MLS mixing ratios (a-c) and OMPS-LP occurrence frequency versus MLS temperature (d-f), both at NLC altitude (determined from OMPS-LP measurements). From a) to c): data for 65°S - 70°S, 70°S - 75°S and 75°S - 80°S. The measurements for 2023/24 are highlighted with filled circles using a red colour for measurements before January 15 and a green colour afterwards. The same for d) to f) but for OMPS-LP occurrence frequency versus MLS temperature.

One of the most striking features in Figure 8 is the high occurrence frequency for the 2022 season that exceeds the 3 $\sigma$
interval for each of the three latitude bins shown. The temperature during 2022 was low during the entire NLC season, whereas
the H$_2$O mixing ratio was unremarkable and confined within the 1 $\sigma$ interval.

Another approach to compare the NH NLC seasons uses the maximum radiance ratio between the measured and background
limb radiance. These measurements show a peak for the 2024 data during the first days of July that exceed 1 $\sigma$ (80 - 85°N) and
3 $\sigma$ (70 - 75°N), respectively (Figure S9). The 2022 NH NLC season does not stand out any more and is well confined within
the 1 $\sigma$ interval between 85 - 75°N.

## 4 Discussion

The massive eruption of the Hunga volcano in January 2022 injected volcanic water vapour into the middle atmosphere that
eventually reached the upper mesosphere. There, it increased the water vapour mixing ratios by approximately 1 ppmv at 70°S
- 75°S beginning in January 2024 at NLC altitude and is lifted to the mesopause region in the NH summer in 2024. Fleming





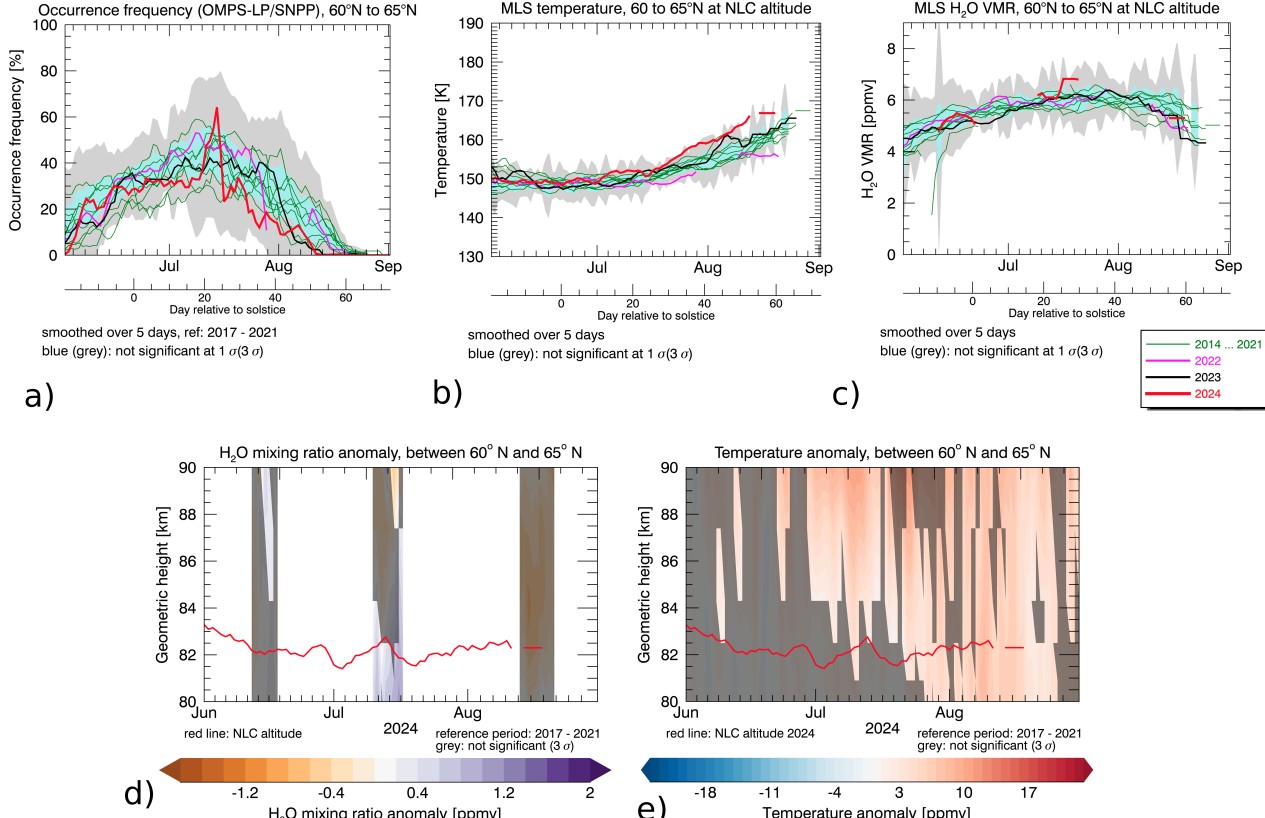

**Figure 7.** A comparison of the OMPS-LP occurrence frequency, MLS temperature, MLS $H_2O$ mixing ratio, the MLS $H_2O$ mixing ratio anomaly and the MLS temperature anomaly between 60 - 65°N. The MLS $H_2O$ mixing ratio and temperature data are shown as anomalies with a reference period of 2017 - 2021 and areas that are not significant to 3 times the standard deviation of the reference period are shaded in grey. The daily and zonally averaged altitude of the NLCs (from OMPS data) is plotted over the MLS data as a red line in panels d) and e).

et al. (2024) speculated that the additional water vapour from Hunga could impact NLCs. Similarly, a lidar station in Argentina (53.8°S) reported the most NLC detections since the beginning of their measurements in 2017 for the recent 2023/24 season (Kaifler et al. (2024)). They speculated whether additional water vapour emission from rocket engine exhausts or the Hunga eruption might be responsible.

The amount of water vapour in the mesosphere depends on water vapour transport, chemical processes (mainly photolysis and methane oxidation) and the solar cycle. The amplitude of the latter effect is about 0.1 ppmv near the mesopause at 0.001 hPa (Lee et al., 2024). Lübken et al. (2018) modeled the impact of a $H_2O$ increase of approximately 1.5 ppmv on the occurrence frequency and the brightness of NLCs. They used the Leibniz Institute Middle Atmosphere Model (LIMA) to simulate an atmospheric background with different $H_2O$ content and the Mesospheric Ice Microphysics And tranSport model (MIMAS)



**Figure 8.** An overview of the OMPS-LP occurrence frequency (left), the MLS temperature (middle) and the MLS $H_2O$ mixing ratio (right) for 65 - 70°N (top), 70 - 75°N (middle) and 75 - 80°N (bottom) during the NH 2024 NLC season. Color schemes and significance intervals are chosen similar to Figure 7.

to simulate the mesospheric ice layer. They reported an increase in both NLC brightness and occurrence frequency due to the additional $H_2O$ in their model. Lee et al. (2024) used data from the Aeronomy of Ice in the Mesosphere (AIM)/Cloud Imaging and Particle Size (CIPS) experiment and Himawari-8/Advanced Himawari Imager (AHI) measurements to investigate the sensitivity of mesospheric clouds to temperature and water vapour content changes in the mesopause region. They supported



the previous argument that water vapour could significantly enhance NLC visibility. Another model study was performed by Yu et al. (2023) using WACCM6 to simulate atmospheric temperature and water vapour contents that were used as input

parameters for a 0-d PMC model. They found that in polar regions (that are already cold enough for NLCs during summer time) mesospheric clouds are more sensitive to water vapour than changes in temperature.

The slight increase in occurrence frequency since mid-January 2024 between 70°S - 75°S compared to the 2016/17 to 2020/21 average could therefore potentially stem from the additional volcanic water vapour. In this case it would be a weak signal and difficult to correlate with the volcanic eruption. The additional Hunga $H_2O$ could also potentially be responsible

for the peak in occurrence frequency in mid-July 2024 in the NH that coincided with an increase in $H_2O$ mixing ratio at NLC altitude. The subsequent decline in occurrence frequency was measured during a period of unusually high temperature that potentially hindered the formation of ice particles. Unfortunately, the MLS instrument has a coarse vertical resolution in the upper mesopause so that an exact correlation between occurrence frequency from OMPS-LP and MLS $H_2O$ and temperature data is limited. One may speculate, whether the comparably high polar summer mesopause temperatures during the second

half of the NH 2024 NLC season might be related to the 2022 eruption of the Hunga volcano. Model simulations by Wallis et al. (2023) showed that massive volcanic eruptions may indeed cause a significant increase in polar summer mesopause temperature, but mainly during the NLC season following the eruption. The temperature pertubations in the SH mesopause decrease significantly during the first two years after the eruption (their Figure 6). For these reasons, it appears unlikely that the high polar summer mesopause temperatures during the NH 2024 season are caused by the Hunga eruption.

The sensitivity of the NLC brightness to the $H_2O$ mixing ratio is well known (Thomas et al., 1989). An increase in available water vapour could result in larger and more particles. Their scattered radiance (i.e. radiant flux per unit angle and area) depends on the effective particle radius. Thus a larger number of bigger ice particles due to additional water vapour should result in brighter NLCs. The sources of additional water vapour could also be human-made. An NLC outbreak observed in June 1963 in Tucson, for example, is speculated to be caused by the water injection from a rocket launch (Russell III et al., 2014). Similarly,

the final launch of the space shuttle was associated with unusually bright NLCs (Stevens et al., 2012).

Lee et al. (2024) reported that the maximum in water vapour mixing ratios at 0.01 hPa and 0.02 hPa in MLS $H_2O$ climatology (2005 - 2021) appears approximately 30 - 40 days after summer solstice. Comparing this to Figure 3b indicates that the $H_2O$ amount at NLC altitude, i.e. above the region described in Lee et al. (2024), was increased approximately during this time period due to the additional volcanic water vapour.

The date of the first reported observation was June 8, 1885 as mentioned in Thomas et al. (1989), i.e. slightly less than two years after the Krakatau eruption on August 27, 1883 (Leslie, 1885; Schröder, 1999). Some records estimate an injection height of 26 km (Self, 1992) up to 40 km (Francis and Self, 1983) for the Krakatau eruption in late August 1883 and a water vapour amount of approximately 100 - 200 Tg (Thomas et al., 1989) up to 500 Tg (Joshi and Jones, 2009), which is comparable with the Hunga overshooting height of 57 km and the estimated 150 Tg $H_2O$ mass for the Hunga event. Using Krakatau as a test

case, it would be plausible that the Hunga water vapour would also need two years to reach the NLC region via the residual mean meridional circulation in the middle atmosphere.



## 5    Conclusions

This study showed that the H$_2$O anomaly from the 2022 Hunga eruption reached the SH summer polar upper mesosphere between 70°S - 80°S during the SH NLC season 2023/24. There, it increased the H$_2$O mixing ratio by approximately 1 ppmv up to 83 km in January 2024. No clear impact from the additional water vapour was detected in the occurrence frequency for the 2023/24 SH NLC season. However, a slight increase in NLC occurrence frequency from mid-January to February compared to previous seasons was observed from OMPS-LP measurements. This could potentially indicate a weak signal of the Hunga eruption that impacted NLC properties two years after the eruption, analogous to the Krakatau event in 1883. The additional water vapour from the Hunga eruption was also transported to the upper polar mesosphere region during the NH summer 2024. The transport of the water vapour is, however, more difficult to track, because the H$_2$O channel of the MLS instrument has not been continually operated any more since April 29, 2024. Even though the current solar maximum would be expected to lower the number of NLCs, there were numerous reports of bright NLC sightings during the first half of the NH 2024 NLC season (spaceweather.com, 2024). There is, however, also no clear Hunga signal present in the occurrence frequencies. MLS data indicate an anomalously high temperature during the second half of the 2024 season at NLC altitude, which might have hindered the formation of ice particles. To summarise, the aftermath of the Hunga eruption offered new insights into the transport of water vapour in the middle atmosphere and the link between volcanic eruptions and noctilucent clouds. Particularly noteworthy is the roughly 2-year lag between the eruption and the time the H$_2$O anomaly reached the polar summer mesopause region.

*Data availability.* The NASA MLS level 2 version 5 H$_2$O data (Lambert and Livesey, 2020) is available on the EarthData GES DISC center (https://disc.gsfc.nasa.gov/datasets/ML2H2O_005/summary?keywords=MLS%20water%20vapour). NASA level 2 version 5 temperatures (Schwartz et al., 2020b) and geopotential heights (Schwartz et al., 2020a) are available at the same website (https://disc.gsfc.nasa.gov/datasets/ML2T_005/summary?keywords=MLS%20temperature and https://disc.gsfc.nasa.gov/datasets/ML2GPH_005/summary?keywords=MLS%20geopotential%20height%20v5%20level%202, respectively). OMPS-LP data was accessed through the https://sbuv.gsfc.nasa.gov/pmc/ompslp/ website.

*Author contributions.* SW and CvS designed the project. MD provided the OMPS-LP NLC detection data set. SW processed the MLS data and the OMPS-LP data. All authors discussed the results and contributed to the writing of the manuscript.

*Competing interests.* The authors declare no competing interests.



*Acknowledgements.* This study is part of the research unit FOR 2820 VolImpact (Grant 398006378) and funded by the German Research Foundation (DFG) within the project VolDyn.



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
