# Peer review of "Has the 2022 Hunga eruption impacted the noctilucent cloud season in 2023/24 and 2024?"

_EGUsphere, 2024_

## Referee Comment (RC2)

Review for manuscript titled "Has the 2022 Hunga eruption impacted the noctilucent cloud season in 2023/24 and 2024?" by S. Wallis, M. DeLand, and C. von Savigny

The manuscript tries to answer the question whether the water vapour injected by the Hunga Tonga-Hunga Ha'apai eruption in 2022 had an impact on the noctilucent cloud (NLC) occurrence in 2023-2024. To answer this question, authors show observations from MLS and OMPS-LP instruments for both hemispheres. Authors also compare observations for 2024 to several previous years to highlight the difference. Despite an increased water vapour mixing ratio in both hemispheres were shown, it was not possible to find a significant change in NLC occurrence frequency. Based on the results shown, authors conclude that it took roughly 2 years for the water vapour injected by the Hunga Tonga-Hunga Ha'apai eruption to reach the polar summer mesopause.

The article is well written and presents results relevant to the journal of Atmospheric Chemistry and Physics. However, there are some minor issues that authors could fix to strengthen the article. I therefore recommend a minor revision that should not be difficult for the authors to perform. Comments are summarized below.

**General comments:**

Throughout the text, authors use "SH NLC season 2023/2024" but "NH NLC 2024 season". It would look better if authors could homogenize it. For example, the year always follows the word season or the opposite.

Figures 2-8 state years 2013/2014 to 2020/2021 in the legend, but the average is shown for 2016/2017 – 2020/2021 referred to as reference period in Section 2.1. It is unclear why one would show more years than needed to calculate the average.

**Minor comments:**

**Abstract**

Line 9: abbreviation NLC is used without being described.

Line 11: authors could consider replacing "seem to" with something more solid like "based on analysis performed in the study, we show/believe/demonstrate/assume"

**Section 3**

Figure 2: please consider adding years to the panel b. This is because many different years are shown in panel a, and it would be easier for the reader to understand which of those years you show in panel b.

Line 107: does occurrence frequency exceed 3 std at the end of February or January?

Line 130: this is the first time seasons 2013/2014 to 2022/23 are mentioned. Please see my general comment on how this is related to the reference period mentioned in Section 2.1.

Line 136-138: based on Figure 6, it does not look like the NLC occurrence frequency in 2023/24 is always higher than in previous years.

Line 139: "Similar scatter plots for latitudes 60°S - 65°S and 65°S - 70°S are shown in Figure S7...", but latitude band 65 – 70 S is already shown in Figure 6, right?

Line 143: consider adding "in the NH" after "is also detected"

Line 151: consider adding "that could explain the decline in NLC occurrence in Figure 7a" after "compared to the mean average". Otherwise, this assumption is only mentioned in the conclusion.

Line 152: please add that the limited vertical resolution of MLS was already mentioned in Section 2.

**Section 4**

Line 206/209: could high $H_2O$ amount be explained by the time of the year when maximum climatological values take place and not be a result of volcanic contribution?

---

## Author Comment (AC1)

**Comments from Anonymous Referee #1**

General comments:

In the present paper, the authors investigate a potential link between the 2022 Hunga Tonga eruption and noctilucent clouds (NLC) activity in the southern and northern hemisphere. The authors have used Microwave Limb Sounder (MLS) measurements of water vapor and temperature in the mesopause region to analyze the development of the mesopause environment and how it has reacted on the 2022 Hunga Tonga eruption. Also, the authors have used Ozone Mapping and Profiling Suite - Limb Profiler (OMPS-LP) measurements to obtain information on NLC activity from 2013 to 2024 in both the SH and the NH. The authors have found a slight increase in water vapor mixing ratio in January-February 2024 by about 1 ppmv between 70°S - 80°S up to an altitude of 83 km. However, no clear signal was observed for the NLC occurrence frequency in the analyzed space and time domains. At the same time, the authors speculate that this slight increase in the H2O amount in the beginning of 2024 in the SH could potentially be caused by the additional water vapour from the 2022 Hunga Tonga massive eruption. Besides, the authors have found a slight increase in the water vapour in the polar summer mesopause region in the NH during the 2024 NLC season. At the same time, the anomalous warming in the NH mesopause during the second half of the 2024 NLC season has been observed that has hindered the NLC formation, and thus masking a potential link between the 2022 Hunga Tunga eruption and the 2024 NLC activity in the NH.

I have found the present paper to be very interesting to the atmospheric community. I recommend the present paper for publication after minor revisions which are outlined below.

We would like to thank the reviewer for taking the time to assess our manuscript and think that their comments resulted in its improvement. We will address all of the reviewer's comments in the following paragraphs.

Specific comments:

Line 11: "To summarise, the volcanic water vapour seems to need two years to reach the summer polar mesopause region."

Please make it clearer here that two years are needed for the volcanic water vapour to reach the summer polar mesopause region from the lower mesosphere.

Thank you for raising our awareness to this aspect. We are, however, hesitant to provide a specific atmospheric region as the starting point for the $H_2O$ transport. As you correctly mentioned, there are reports of the visible plume reaching 57 km altitude (Proud et al., 2022). Nevertheless, there are also indications that the plume only shortly reached this overshooting height before it collapsed. MLS data and model simulations by Niemeier et al. (2023) indicate, that the main part of the plume descended in the first two weeks and remains between 20 - 40 hPa until October 2022. Because of this complex behavior we rephrased the sentence to "To summarise, based on analysis performed in the study, we show that the volcanic water vapour needs two years to reach the summer polar mesopause region".

Reference:

Simon R. Proud, Andrew T. Prata & Simeon Schmauß (2022),The January 2022 eruption of Hunga Tonga-Hunga Ha'apai volcano reached the mesosphere. Science378,554-557. DOI:10.1126/science.abo4076

Niemeier, U., Wallis, S., Timmreck, C., van Pham, T., & von Savigny, C. (2023). How the Hunga Tonga—Hunga Ha'apai water vapor cloud impacts its transport through the stratosphere: Dynamical and radiative effects. Geophysical Research Letters, 50, e2023GL106482. https://doi.org/10.1029/2023GL106482

Lines 23-24: "Two years later, in June 1885, first sightings of noctilucent clouds were reported (Backhouse, 1885; Leslie, 1885; Schröder, 1999)."

Please add here the paper by Tseraskii (1887) who observed, photographed and estimated the NLC altitude for the first time already in June 1885.

We added the paper by Tseraskii (1887) as suggested.

Lines 185-186: "They found that in polar regions (that are already cold enough for NLCs during summer time) mesospheric clouds are more sensitive to water vapour than changes in temperature."

Here it is worth mentioning the paper by Pertsev et al. (2014) which clearly demonstrated the sensitivity of NLC to the relative humidity of the mesopause region. At the same time, it should be mentioned that Dalin et al. (2023) showed that a combination of lower mesopause temperature and water vapor mixing ratio maximum at middle latitudes was the main reason for frequent and widespread occurrences of NLC seen around the globe at middle latitudes in the 2020 summer.

We agree and added "The sensitivity of NLCs to the relative humidity of the mesopause region was also confirmed by Pertsev et al. (2014). Moreover, Dalin et al. (2023) showed that a combination of low mesopause temperature and a maximum in water vapor mixing ratios was the main reason for the frequent and widespread occurrences of NLCs seen at NH mid-latitudes in the summer of 2020." to the text.

Lines 204-205: "Similarly, the final launch of the space shuttle was associated with unusually bright NLCs (Stevens et al., 2012)."

Here it is worth mentioning the paper by Dalin et al. (2013) which clearly demonstrated the direct formation of NLC in the rocket exhaust trail.

We agree and added "and Dalin et al. (2013) even demonstrated the direct formation of NLCs in the exhaust trails of Soyuz rockets." to the text.

Additional references:

Dalin, P., H. Suzuki, N. Pertsev, V. Perminov, N. Shevchuk, E. Tsimerinov, M. Zalcik, J. Brausch, T. McEwan, I. McEachran, M. Connors, I. Schofield, A. Dubietis, K. Černis, A. Zadorozhny, A. Solodovnik, D. Lifatova, J. Grønne, O. Hansen, H. Andersen, D. Melnikov, A. Manevich, N. Gusev, V. Romejko: The strong activity of noctilucent clouds at middle latitudes in 2020. Polar Science, 35, 100920, https://doi.org/10.1016/j.polar.2022.100920, 2023.

Dalin, P., Perminov, V., Pertsev, N., Dubietis, A., Zadorozhny, A., Smirnov, A., Mezentsev, A., Frandsen, S., Grönne, J., Hansen, O., Andersen, H., McEachran, I., McEwan, T., Rowlands, J., Meyerdierks, H., Zalcik, M., Connors, M., Schofield, I., Veselovsky, I.: Optical studies of rocket exhaust trails and artificial noctilucent clouds produced by Soyuz rocket launches, JGR-Atmospheres, 118, 14, 7850-7863, https://doi:10.1002/jgrd.50549, 2013.

Pertsev, N., Dalin, P., Perminov, V., Romejko, V., Dubietis, A., Balčiunas, R., et al.: Noctilucent clouds observed from the ground: sensitivity to mesospheric parameters and long term time series. Earth, Planets and Space, 66(1), 1–9, https://doi.org/10.1186/1880 5981 66 98, 2014.

Tseraskii, V. K.: Astronomichesky fotometr i ego prilozhenia (Astronomical photometer and its applications). Doctoral Dissertation, Mathematical Proceedings, XIII, Section 21, 626–631, 1887 (in Russian).

We would like to thank Anonymous Referee #1 for their time and for providing comments on our manuscript.

---

## Author Comment (AC2)

**Comments from Anonymous Referee #3**

Review for manuscript titled "Has the 2022 Hunga eruption impacted the noctilucent cloud season in 2023/24 and 2024?" by S. Wallis, M. DeLand, and C. von Savigny
The manuscript tries to answer the question whether the water vapour injected by the Hunga Tonga-Hunga Ha'apai eruption in 2022 had an impact on the noctilucent cloud (NLC) occurrence in 2023-2024. To answer this question, authors show observations from MLS and OMPS-LP instruments for both hemispheres. Authors also compare observations for 2024 to several previous years to highlight the difference. Despite an increased water vapour mixing ratio in both hemispheres were shown, it was not possible to find a significant change in NLC occurrence frequency. Based on the results shown, authors conclude that it took roughly 2 years for the water vapour injected by the Hunga Tonga-Hunga Ha'apai eruption to reach the polar summer mesopause.
The article is well written and presents results relevant to the journal of Atmospheric Chemistry and Physics. However, there are some minor issues that authors could fix to strengthen the article. I therefore recommend a minor revision that should not be difficult for the authors to perform. Comments are summarized below.

We would like to thank the reviewer for taking the time to revise our manuscript and agree that her/ his comments resulted in its improvement. We will address all of the reviewer's comments below.

General comments:
Throughout the text, authors use "SH NLC season 2023/2024" but "NH NLC 2024 season". It would look better if authors could homogenize it. For example, the year always follows the word season or the opposite.

We agree and changed the relevant parts of the text to make the phrasing consistent.

Figures 2-8 state years 2013/2014 to 2020/2021 in the legend, but the average is shown for 2016/2017 – 2020/2021 referred to as reference period in Section 2.1. It is unclear why one would show more years than needed to calculate the average.

As suggested, we revised Figure 2 -8 and now only show data starting from 2016/17 for the SH and 2017 for the NH.

Minor comments:
Abstract
Line 9: abbreviation NLC is used without being described.

The abbreviation NLC is replaced by "noctilucent cloud".

Line 11: authors could consider replacing "seem to" with something more solid like "based on analysis performed in the study, we show/believe/demonstrate/assume"

We agree and replaced "seem to" with "based on analysis performed in the study, we show that" as suggested.

Section 3
Figure 2: please consider adding years to the panel b. This is because many different years are shown in panel a, and it would be easier for the reader to understand which of those years you show in panel b.

A date was added to Figure 2b to improve the clarity of the plot.

Line 107: does occurrence frequency exceed 3 std at the end of February or January?

We would like to thank the reviewer for pointing out this mistake. We corrected "February" to "January."

Line 130: this is the first time seasons 2013/2014 to 2022/23 are mentioned. Please see my general comment on how this is related to the reference period mentioned in Section 2.1.

We changed all of the Figures to only show data starting in 2016/17 for the SH and 2017 for the NH and corrected their description in the text accordingly.

Line 136-138: based on Figure 6, it does not look like the NLC occurrence frequency in 2023/24 is always higher than in previous years.

We agree and added "on average".

Line 139: "Similar scatter plots for latitudes 60°S - 65°S and 65°S - 70°S are shown in Figure S7...", but latitude band 65 – 70 S is already shown in Figure 6, right?

We thank the reviewer for drawing our intention to that and removed Figure S7a.

Line 143: consider adding "in the NH" after "is also detected"

We agree and added "in the NH" to make the sentence clearer.

Line 151: consider adding "that could explain the decline in NLC occurrence in Figure 7a" after "compared to the mean average". Otherwise, this assumption is only mentioned in the conclusion.

We agree and added the phrase as suggested.

Line 152: please add that the limited vertical resolution of MLS was already mentioned in Section 2.

The phrase "as discussed in Section 2" was added.

Section 4
Line 206/209: could high $H_2O$ amount be explained by the time of the year when maximum climatological values take place and not be a result of volcanic contribution?

In order to account for the seasonal variation in mesospheric $H_2O$ mixing ratios, we calculated the anomaly compared to a five year reference period, i.e. the 5-year multi-annual monthly means were subtracted from the respective current monthly mean (Figure 1 and Figures S1 – S3). Only Figures 2b and 7d show anomalies based on daily means. Nevertheless, we agree that a variation in the seasonal $H_2O$ content could result in positive or negative $H_2O$ anomalies. Figure S1-S3 show the monthly $H_2O$ mixing ratio anomalies from January 2022 to August 2024 and highlight the areas, that exceed more than 3 times the standard deviation of the 2017 – 2021 reference period. These areas seemed to be well connected and appear to be due to the volcanic $H_2O$ emission. Therefore, we focus on the contribution of the Hunga volcano to the $H_2O$ budget in the mesosphere as we think that this will be the dominant cause for the anomalies.

We would like to thank Anonymous Referee #3 for their time and for providing comments on our manuscript.

---

## Author Response (AR2)

Dear Matthias Tesche,

thank you very much for taking the time to comment on our manuscript. We changed the colour scheme to make the plots suitable for readers with colour vision deficiencies. Moreover, we removed the information on smoothing, reference period, and significance in the plots themselves and provide them in the captions. We agree that these modifications make the manuscript more accessible and improve the clarity of the plots.